# Emotional and Work-Related Factors in the Self-Assessment of Work Ability among Italian Healthcare Workers

**DOI:** 10.3390/healthcare12171731

**Published:** 2024-08-30

**Authors:** Nicola Magnavita, Igor Meraglia, Carlo Chiorri

**Affiliations:** 1Department of Life Sciences and Public Health, Università Cattolica del Sacro Cuore, 00168 Roma, Italy; igor.meraglia01@icatt.it; 2Department of Educational Sciences, University of Genova, 16126 Genova, Italy; carlo.chiorri@unige.it

**Keywords:** disability management, medical surveillance, health promotion, epidemiology, effort–reward imbalance, overcommitment, social support, job satisfaction, anxiety, depression, happiness

## Abstract

The Work Ability Index (WAI) is the most commonly used tool for evaluating work capacity. Self-assessments made by workers can be influenced by various occupational and emotional factors. We wanted to study the association of work-related factors, such as work annoyance, stress, overcommitment, job satisfaction, social support, and emotional factors, such as anxiety, depression, and happiness, with work ability, in a sample of 490 healthcare workers from an Italian public health company. A principal component analysis indicated the presence of two components of the WAI questionnaire; the first expresses “subjectively estimated work ability” (SEWA), and the second refers to “ill-health-related work ability” (IHRWA). Using stepwise multiple hierarchical linear regression, we identified the factors that best predicted the total score on the WAI and on the two components. The total score was negatively predicted by anxiety, depression, a lack of happiness, low job satisfaction, overcommitment, and work annoyance. Age, being female, anxiety, and occupational stress were associated with a reduction in the IHRWA component score, while overcommitment, work annoyance, a lack of social support, depression, and a lack of happiness were negatively associated with the SEWA component. These results can help interpret those of epidemiological studies and provide guidance on ways to improve work ability.

## 1. Introduction

The concept of work ability is one of the most important in occupational medicine, and the articulation of its meaning has developed progressively over the last 40 years. According to Ilmarinen and Tuomi [1], work ability should be based on four elements: (1) health and functional capacities; (2) knowledge and skills; (3) values, attitudes, and motivation; and (4) work situation/work demands. Tengland [2] observed that occupational medicine requires two definitions of work ability: one for professions requiring specialized education or training, and another for jobs that most people can perform with a little amount of experience. Assuming that the tasks are reasonable, and the work environment is acceptable, the first definition of work ability refers to having occupational competence, the health necessary for competence, and the occupational virtues necessary for handling the job’s tasks. In the second sense of the meaning, being able to perform work implies possessing the physical well-being, a basic general competency, and pertinent professional attributes needed to oversee the job. To date, there is no unified, comprehensive understanding of the precise characteristics and scope of work ability in the scientific literature. Work ability is a relational concept that emerges from the interaction of several dimensions acting on various ecological levels [3]. The concept of work ability has also become more dynamic over time as work environments have evolved.

To evaluate work ability, several questionnaire measures are available [4,5]. The Work Ability Index (WAI) [6] is likely the most widely used [7,8]. The questionnaire comprises seven components, numbered WAI 1 through WAI 7, that collectively aim to gauge an individual’s work ability. The seven components are characterized as follows: WAI1, present work ability in relation to lifetime best; WAI2, work ability compared to job demands; WAI3, total number of illnesses currently being treated by a doctor; WAI4, estimated illness-related job impairment; WAI5, sick leave taken within the previous year; WAI6, individual projection of work capacity in two years; and WAI7, mental resources.

According to the authors’ indications [1,6,7] and most field studies [9,10,11,12,13,14,15,16,17,18,19,20,21,22,23,24,25,26,27], the seven dimensions combine to form a single total score, ranging from 7 to 49. Work ability can be categorized as excellent (WAI score of 44–49), good (WAI score of 37–43), moderate (WAI score of 28–36), and poor (WAI score of ≤27). Ebener and Hasselhorn [28] proposed a conceptualization of the WAI as a formative measure, which comprises a distinct constellation of well-selected items with minimal common variance among them. Some psychometric studies supporting the validity of the one-factor structure have observed that a greater amount of information can be obtained if two components, defined as “subjectively estimated work ability” (SEWA) and “ill-health-related work ability” (IHRWA), are considered separately [29,30,31,32]. The first component (composed of indicators 1, 2, 6, and 7) includes a self-evaluation of one’s working capacity in relation to one’s lifetime best capacity, as well as an assessment of one’s resources and effectiveness in relation to the psychological and physical demands of one’s line of work; the second factor (indicators 3, 4, and 5) gathers data on medically diagnosed illnesses, their effects on productivity, and the number of sick leave absences. Cadiz et al. [33] suggested a structure with two WAI subdimensions, refining the labeling to distinguish between “subjective” and “objective” work ability. However, one could question the validity of labeling a list of self-reported illnesses and medical diagnoses, produced through social interaction, processed by an individual’s mind, and then self-reported in a survey, as ‘objective’. The existence of a gap between the subjective evaluation of one’s work ability and reality is demonstrated by the fact that, in some studies, the WAI is inversely proportional to sickness absences; males have a WAI on average higher than women, but the two sexes have the same number of absences [34]. The paradox can only be explained by admitting that cultural and social factors act on subjective evaluation. In any case, the problem of distinguishing a real reduction in work ability from a personal underestimation of one’s abilities remains unsolved.

The WAI is a tool widely used in occupational medicine to enable workers to self-assess their working capacity. This assessment can be useful for different purposes. It helps clinicians and researchers to assess work abilities during health exams and workplace surveys in clinical occupational health and research. It also evaluates suitability for the job task and suggests environmental and organizational changes, as well as health promotion measures, to empower the worker to apply themselves successfully to the job. Research has shown that the WAI can predict several outcomes relevant to occupational health (and other disciplines), such as long-term sick leave [35,36,37,38], work nonparticipation [39], early leave from work [40,41], death and disability [42,43], willingness to return to work after SARS-CoV-2 infection [44], or the impact of post-COVID symptoms [45,46,47], among others. A longitudinal study has shown that poor work ability exposes workers to an increased risk of violence at work [48]. Many large epidemiological investigations have used the WAI. At baseline, around 30,000 people from ten European countries participated in the Nurses’ Early Exit study, and over 9000 from eight countries responded at follow-up [49,50]. In Sweden, the Work, Health, and Retirement Study (WHRS) covered a wide sample of the working population [51]. About 7000 healthcare workers were studied in Friuli, an Italian region [52]. Large-scale longitudinal investigations have also been conducted in Japan [53] and China [54]. It has been applied to evaluate the state of a working population before medical examination [34], the conditions of patients at the end of a therapeutic path [55,56,57], or even improvement after a disability management program [58,59,60]. The most common use, however, is to support an occupational doctor’s visit during worker health surveillance. Healthcare workers are the most frequently studied category, but it has been applied to many categories of workers. Given the questionnaire’s widespread use, in-depth knowledge of the instrument is required. For the occupational doctor, the disability manager, or the psychologist, it is important to know which work factors could have influenced the measured level of work ability, with the purpose of intervening in the environment or in the organization of work. At the same time, it is necessary to know whether certain emotional factors have affected the worker’s self-evaluation and correct these factors with health promotion interventions, to improve worker integration and increase job satisfaction.

Numerous factors can influence one’s self-assessment of work ability, which has been known for some time. Many organizational, occupational, or personal factors were associated with WAI scores in cross-sectional [61,62,63,64,65,66,67] and in longitudinal studies [68,69]. For example, Mazloumi et al. [61] believed that maintaining work ability was mostly dependent on several elements, including social support, skill discretion, job strain, and job instability. In a cross-sectional study, Bethge et al. [65] observed that high job strain, due to high demand and low control, as well as an effort–reward imbalance, independently explained low levels of work ability. Gajewski et al. [67] demonstrated that a number of factors, including age, psychosocial stress, work demands, subclinical depression, and burnout symptoms, were associated with low WAI scores in a sample of German workers of different occupations. Martinez et al. [68], in a longitudinal study on healthcare workers, demonstrated that work stressors (job control, social support, effort–reward imbalance, and overcommitment) negatively affected work ability over time. In contrast, Andrade et al. [69], studying a sample of public servants with stability and working from home during the COVID-19 pandemic, did not observe significant changes in work ability that could be ascribed to the pandemic.

The available studies have generally examined a small number of factors, almost always with a cross-sectional design. Although cross-sectional studies cannot indicate the direction of associations, it is plausible to believe that occupational factors are responsible for work ability, while the inverse relationship is unlikely. For emotional factors, such as anxiety, depression, or happiness, the absolute lack of longitudinal studies makes it impossible to understand whether the observed association indicates that work ability influences mental health or, as seems more likely, that mental health status influences the self-evaluation of work ability when completing the questionnaire. In addition, although several studies have noted that WAI scores may be associated with occupational and emotional factors, there is no study that has considered individual and work-related factors simultaneously, to obtain an evaluation of the relative weight of each variable on the resulting WAI score. This gap in the literature must be filled.

In this study, our aim was to verify which occupational and psychological factors are associated with WAI scores and investigate their relationship and strength of association with WAI scores, using a stepwise multiple hierarchical linear regression. Specifically, we first wanted to test how occupational factors predicted WAI scores while controlling for age and sex. Second, we tested whether the addition of emotional factors (anxiety, depression, and happiness) to the model increased its predictive ability. It seemed logical to us to study the occupational factors first, because they are the ones that can be modified through changes in the work organization, and to then study the emotional factors, which would require individualized interventions for workers. We used a progressive variable selection method to obtain an idea of the relative importance of the different factors.

Knowledge of organizational and personal factors that can influence work ability can be of great use to occupational physicians, managers, and safety operators, since a high work ability is associated with workers’ well-being, satisfaction, quality of life, and productivity.

We deemed it appropriate to study the workers of a public healthcare company, since healthcare workers are the category most frequently investigated using the WAI and since, in a public company, the risk of job insecurity, which could be inversely associated with WAI score [70], is minimal. Moreover, public workers enjoy a national employment contract [71], which establishes homogeneous working conditions.

The hypotheses that, based on the literature, we wanted to test were as follows:Work ability decreases with age;Work ability is greater in males than in females;Occupational factors (work annoyance, effort–reward imbalance, overcommitment, social support, and job satisfaction) are associated with work ability;Emotional factors (anxiety, depression, and happiness) are associated with work ability.

Furthermore, we wanted to understand whether the variables indicated above are associated with the total score of the WAI or with that of its components. Finally, we wanted to obtain an idea of the relative weight of each variable.

## 2. Materials and Methods

### 2.1. Population

European Directives and national laws in Italy impose a health surveillance program on workers exposed to professional risks, which involves periodic workplace visits and a doctor’s assessment of the worker’s suitability for the job, taking professional risks into account. Optional health promotion programs can complement this legally mandated preventive activity. The health surveillance of workers, which includes occupational risk prevention and health promotion activities, is mandatory in all workplaces where workers are exposed to occupational risks. The data collected during this surveillance are confidential and cannot be disclosed. The results relevant to health and safety are communicated in an anonymous collective form to the employer, the risk prevention service, and the workers’ representatives and may be the subject of scientific communications.

The health surveillance service of a public health company invited its visiting employees to participate in a health promotion activity that included an evaluation of their work ability. The company’s occupational health physician invited workers who agreed to participate to complete a questionnaire immediately before their periodic medical examination. The questionnaire measured, in addition to work ability, work-related stress, social support, overcommitment, job satisfaction, work annoyance, anxiety, depression, and happiness. Participation was free, not incentivized, and independent of work suitability assessment; 490 of the 535 workers (91.6%) agreed to participate. Among the reasons for non-participation, workers mainly indicated the lack of time to fill out the questionnaire. Only two of them declared that they considered the health promotion activity useless. The sample was very homogeneous in terms of geographical and linguistic origin; 486 (99.2%) were born and lived in Italy.

We conducted the research in accordance with the Declaration of Helsinki. The participants gave their informed consent by signing the personal health document, consented to the analysis of their personal data, and agreed to the dissemination of the results in an anonymous aggregate form in accordance with the occupational medicine confidentiality principles and the International Commission on Occupational Health (ICOH) code of ethics [72]. The Catholic University Ethics Committee granted ethics approval (ID 2896). The cross-sectional nature of the study prevented the imputation of missing data, and the findings relied solely on completed survey responses.

### 2.2. Questionnaire

Before their periodic medical examination, workers self-assessed their working capacity using the Italian version [73] of the WAI [5]. The WAI consists of a set of inquiries that consider the worker’s resources, health, and the physical and mental demands of their job. In this study, we used the unitary score (WAItot), which most of the literature uses, and we also used the two-component score (SEWA and IHRWA), according to the principal component analysis.

The reliability of the questionnaire in this present study (Cronbach’s alpha) was 0.685 and therefore within the range of values (0.54 to 0.83) observed in previous studies performed with this tool [31,32,74,75,76,77].

The workers were also invited to fill out other questionnaires to evaluate their occupational conditions. The Work Annoyance Scale (WAS) [78] indicates the degree of boredom produced by various working conditions. The questionnaire is made up of nine questions, each starting with “How much does it bother you to…” and proposes a series of work situations. The answers have values between 0 and 10; the overall score, which derives from the sum of the answers, can range from 0 to 90. The Cronbach’s alpha of the questionnaire in this study was 0.839.

Work-related stress was measured with the short form [79] of the Siegrist’s effort/reward imbalance (ERI) questionnaire [80], specifically the Italian version [81]. This tool measures two constructs: effort (3 items) refers to the effort required to work, and reward (7 items) measures the material and immaterial rewards received. A stress measure is calculated as the weighted ratio between the two variables (ERI Stress). A score higher than 1 is considered an expression of a condition of distress due to an imbalance between effort and result. The reliability of the Effort scale was 0.782; the Cronbach’s alpha of the Reward scale was 0.623.

In this study, we measured Overcommitment, a dimension that is often used interchangeably with workaholism to characterize someone who is excessively dedicated to their own work. Previous studies, however, demonstrated that overcommitment, rather than workaholism, is the only factor that is specifically linked to job burnout and has a greater correlation with neuroticism than workaholism, so it may be the real drawback of work drive [82], representing the intrinsic component of the effort–reward model of stress [83]. Overcommitment was measured with six items, rated on a 4-point scale, with a range of possible scores of 6–24. Cronbach’s alpha was 0.775.

Social support, indicating a moderating factor in stress models [84], was measured with a six-item scale rated on a 4-point scale, as per the Italian version of the questionnaire (range 6–24) [81]. Its reliability in this study was 0.796.

Job satisfaction was measured with a single item, taken from the Warr and Cook scale [85], specifically the Italian version [86,87], with scores ranging from 1 to 7.

To measure emotional factors, workers completed the Italian version [88] of the Goldberg Anxiety and Depression Scale (GADS) [89], which is made up of 9 binary questions related to anxiety and 9 relating to depression. One point is awarded for each positive answer. Workers who have a score of 5 points for anxiety and 2 for depression have a 50% chance of suffering from these pathological conditions; the risk increases rapidly as the score increases. The Anxiety scale had a Cronbach’s alpha of 0.84; the Depression scale had an alpha coefficient of 0.795. Happiness was measured using Abdel-Khalek’s 1-item scale [90], graded from 0 to 10.

### 2.3. Statistics

The distribution of the relevant variables was examined using the Shapiro–Wilk and Kolmogorov–Smirnov tests, to see whether it was normal. We followed the general principle of applying nonparametric tests to non-normally distributed variables; however, as reported by Lumley et al. [91], the assumption of normality is only required for small samples, due to the central limit theorem. With sample sizes exceeding 30, as was the case here, violations of the normality assumptions are not problematic, and the use of both parametric and non-parametric tests is appropriate. To test whether the WAI score could be divided into two dimensions, we used a principal component analysis with oblimin rotation and Kaiser normalization of the factors.

We tested the bivariate correlation between the total score of the WAI and that of its components, sex, age, and occupational and emotional variables using Spearman’s rho and Pearson’s *r*, with a two-tailed test for significance.

The predictive value of the occupational or emotional variables on work ability was tested using stepwise multiple hierarchical linear regression models, which were age- and sex-adjusted. Occupational variables were entered in the first block (Model I) and emotional ones in the second (Model II). We applied a stepwise selection method to any group of variables, in order to find the variables that best predicted WAI scores. A probability-of-F-to-enter of ≤0.050 and a probability-of-F-to-remove of ≥0.100 were adopted as stepwise criteria. In the regression, we tested the variables for multicollinearity using the variance inflation factor (VIF), which measures the correlation and strength of correlation between the predictor variables. The value for VIF starts at 1 and has no upper limit. A value between 1 and 5 indicates a moderate correlation between a given independent variable and other independent variables in the model, but this is often not severe enough to require attention.

IBM/SPSS Statistics for Windows, Version 28.0 (IBM Corp.: Armonk, NY, USA) was used for the analyses.

## 3. Results

The workers who participated in the project were on average 44.9 years old (standard deviation s.d. 12.5) and were predominantly female (325, 66.3%). The most represented professional category was nurses (324 workers, 66.1%), followed by doctors (100, 22.4%), technicians (34, 6.9%), and clerks (32, 6.5%).

The average score on the WAI questionnaire was equal to 40.6 ± 5.5 points; the median value was 42, and WAI score was inversely correlated with age (Spearman’s rho = −0.106 *p* < 0.001, Pearson’s *r* = −0.205, *p* < 0.001).

Female workers had a significantly worse WAI score than males (40.15 ± 5.46 in female vs. 41.60 ± 5.56 in male workers, *t*(323,945) = 2.73, *p* = 0.007, *d* = 0,26).

Mean WAI scores showed small differences between the different professional categories (*F*(3, 480) = 4.06, *p* = 0.007, η^2^ = 0.03). Nurses reported the lowest WAI mean score, but the Bonferroni-corrected post hoc tests showed that it was significantly lower only than that of office workers (*t*(480) = 2.98, *p* = 0.018, *d* = 0.55) (Table 1).

The principal component analysis of the seven indicators of the WAI (Table 2) suggested two dimensions, the first (38.2% of the variance) grouping indicators 1, 2, 6 and 7, and the second (18.3% of the variance) grouping indicators 3, 4 and 5. These factors corresponded to the SEWA and IHRWA factors previously reported the literature. The two components were moderately correlated (rho = 0.244, *p* < 0.001; *r* = 0.336, *p* < 0.001).

The total WAI score was significantly correlated with all the variables that we included in this study, both those related to work and those referring to the emotional state of the workers who completed the questionnaire (Table 3). Among work-related factors, WAI was negatively correlated with ERI Stress (rho = −0.376, *p* < 0.001; *r* = −0.393, *p* < 0.001), Work Annoyance (rho = −0.369, *p* < 0.001; *r* = −0.377, *p* < 0.001), and Overcommitment (rho = −0.376, *p* < 0.001; *r* = −0.422, *p* < 0.001), and positively with Social Support (rho = 0.264, *p* < 0.001; *r* = 0.276, *p* < 0.001) and Job Satisfaction (rho = 0.237, *p* < 0.001; *r* = 0.258, *p* < 0.001). Among personal factors, WAI was negatively correlated with Anxiety (rho = −0.488, *p* < 0.001; *r* = −0.490, *p* < 0.001) and Depression (rho = −0.464, *p* < 0.001; *r* = −0.490, *p* < 0.001) and positively with Happiness (rho = 0.372, *p* < 0.001; *r* = 0.396, *p* < 0.001). The SEWA component was not significantly correlated with age but was significantly correlated with sex and with all occupational and emotional variables of interest. The IHRWA component was significantly correlated with all occupational and emotional variables, except Social Support (Table 3).

The results of the regression analyses showed that Model I, which included only age, sex, and occupational factors as predictors, showed that being female, lower levels of overcommitment, work annoyance, and stress, as well as higher levels of satisfaction were significantly associated with the total WAI score (Table 4). This model accounted for 29.3% of variance (as indexed by the adjusted R^2^). The SEWA component was predicted by higher age, lower levels of work annoyance and overcommitment, and higher levels of social support and satisfaction (adjusted R^2^ = 0.327). The IHRWA component was predicted by lower age, being female, and lower levels of stress and overcommitment (adjusted R^2^ = 0.141). In all tested models, the VIF values of the predictors were close to unity, indicating that the degree of correlation between the predictors in the model was not sufficient to influence the outcome.

Model II, which also included emotional factors, revealed that the total WAI score was predicted by younger age, lower levels of work annoyance, overcommitment, anxiety, and depression, and higher levels of satisfaction and happiness (adjusted R^2^ = 0.374). The SEWA component was predicted by lower levels of work annoyance, overcommitment, and depression, and higher levels of social support and happiness (adjusted R^2^ = 0.412). The IHRWA component was predicted by lower age, being female, and lower levels of stress and anxiety (adjusted R^2^ = 0.179)

## 4. Discussion

This study tested which occupational and emotional factors significantly predicted work ability over and above the others. The result of the regression analyses showed that, once adjusted for age and sex, factors of either group were significant predictors of the total WAI score, accounting for over 37% of variance. Specifically, the total WAI score was predicted by younger age, lower levels of work annoyance, overcommitment, anxiety, and depression, and higher levels of satisfaction and happiness. A different pattern of results was found when we used as a criterion in the regression analysis the two components of WAI, i.e., subjectively estimated work ability (SEWA) and ill-health-related work ability (IHRWA). While the SEWA component was significantly predicted by lower levels of work annoyance, overcommitment, and depression, and higher levels of social support and happiness, which accounted for over 40% of variance, the IHRWA component was significantly predicted by lower age, being female, and lower levels of stress and anxiety, which accounted for less than 20% of variance. It is worth noting that the obtained models did not suffer from multicollinearity and were therefore reliable. Multicollinearity, i.e., a strong correlation between the predictor variables, often leads to incorrect results in regression analyses [92,93,94,95].

Compared to previous studies, which had already investigated the effect of the factors included in this work but separately, this study allowed us to test the different contributions of occupational and emotional factors to the self-assessment of work ability simultaneously, not only on the total WAI score, but also on the component scores. Interestingly, the results showed different patterns of results. For example, age was negatively correlated with the total score and the IHRWA component but was positively correlated with the SEWA component. This seemingly paradoxical effect could be attributed to the fact that, while it is true that the prevalence of chronic diseases increases over time and can impact work performance, thereby increasing the HIRWA component and, to a lesser extent, the WAI component, it is also true that older individuals with more experience and higher skills are better at evaluating their ability to solve work-related problems, thereby increasing the SEWA component. This result can help to explain the inconsistency of some previous results, since some studies reported that the WAI score decreases with age [62,67,96,97,98,99,100,101,102], some found no association [103], and some reported that the WAI score increases with age [104]. A longitudinal study conducted on public employees has shown that work ability can increase significantly with age, thus demonstrating that work ability depends more on health and safety, promotion, and preventive activities at the workplace than on the passage of time [105].

In this study, women reported lower total and component scores than men, although with small effect sizes, consistent with previous studies [34,52,106,107,108]. However, in the regression analyses, the effect of sex was significant only for the IHRWA component.

Previous studies [109,110,111] have already reported the association of occupational factors with the process of self-assessment of work ability. According to traditional interpretations, a low work ability is linked to stress. However, some authors have reported that reduced work ability can cause workers to feel stressed [112]. Cross-sectional studies cannot clarify the direction of the association observed between stress and work ability. Only longitudinal studies, such as the one conducted on the Stockholm population between 2010 and 2014, have been able to clarify that poor work ability at the baseline is associated with a higher incidence of future psychological distress compared to good work ability [113].

It should also be noted that stress is a complex phenomenon, whose definition is based on precise theoretical models and the interactions of various components. To the best of our knowledge, there are no studies evaluating the relationships between different components of occupational stress and WAI. We looked at two types of stress in this study: extrinsic job stress, which Siegrist’s model defines as the difference between how hard someone works and how much they are paid, and intrinsic stress, also known as overcommitment. This is because both of these have been shown to be independent predictors of mental health [114]. In this study, we found that stress and overcommitment were moderately correlated, but they differentially predicted the WAI components: while extrinsic stress predicted IHRWA, overcommitment predicted the WAI total score and SEWA.

Furthermore, we entered in the regression models further predictors that previous studies have found to be associated with WAI scores and psychological well-being (or lack thereof) in the workplace, such as social support at work and job satisfaction [68,115,116,117] and work annoyance, i.e., the subjective evaluation of work characteristics as something causing vexation or nuisance [78]. The results showed that all these occupational factors significantly predicted the WAI score and/or its components. Worrying for possible negative working conditions (Annoyance), extrinsic stress (ERI) and intrinsic stress deriving from one’s excessive commitment (Overcommitment) were the variables that most were associated with lower levels of work ability, while Social Support and Satisfaction Derived From Work were associated with higher levels. These results are in agreement with a longitudinal study showing that Social Support increases and Overcommitment decreases the WAI score [68], and with cross-sectional studies showing that ERI is associated with poor work ability [118].

Emotional factors, too, showed moderate bivariate correlations with WAI scores, and could predict the WAI scores over and above occupational factors in the regression models. Anxiety, depression, and low levels of happiness were associated with low WAI values. Depression and low happiness were significant predictors of one’s skills component (SEWA). Conversely, the component associated with poor health (IHRWA) experienced a reduction in anxiety and stress. This indicates that depressed and unhappy workers could evaluate their skills less favorably than they actually are and again provide a low WAI. Similarly, anxious and distressed workers may place greater importance on their health problems than their non-anxious colleagues and, consequently, rate their ability to work more poorly. These results are not surprising, considering the findings of previous studies that showed that depressive symptoms were associated with lower work ability [119]. Individuals with burnout tend to develop depression and a reduced work ability [120]. The association between poor work ability and depression may also be interpreted in the sense that reduced work capacity may induce a state of depression in workers [121]. In women treated for climacteric, an improvement in post-menopausal depressive symptoms was associated with an improvement in WAI scores reported at follow-up [122]. In women who survived breast cancer and returned to work, low work ability was associated with symptoms of anxiety and depression [55], and, in the absence of psychotherapy, mental health and work ability did not show significant improvement one year after returning to work [123]. Conversely, in another prospective study, improved psychiatric symptoms after treatment were associated with an increase in WAI scores at follow-up [124]. The treatment of depression caused by occupational stress achieves the greatest improvements in work ability when psychotherapy treatments include an interpersonal focus on the work environment [125].

Our findings also highlighted that work ability measured by the WAI is a dynamic concept. The questionnaire asks workers to evaluate their health conditions in relation to the tasks required; therefore, the evaluation changes as the work task varies. The level of skills and resources assessed by the worker may depend on the demands of the job. Furthermore, the concept of work ability is multifactorial, since the influence of each of the seven components will vary across the range of scores [126]. The physician using the WAI questionnaire should carefully consider these characteristics of work ability and bear in mind that, although expressed in numerical terms like the percentages of disability, the WAI score is a completely different measure, of a subjective nature. Workers with a low level of mental well-being may underestimate their ability to work. Similarly, individuals who are excessively involved in work may indicate a lower level of work ability than they actually are.

In addition to the occupational and emotional factors we have studied here, self-evaluation can be influenced by social and cultural factors. An indirect demonstration of the role of these factors comes from the inconsistent results obtained from psychometric studies of WAI. Most of the authors confirmed the unifactorial structure suggested by the creators of the questionnaire [127] or indicated better properties in relation to a two-factor structure [29,30,128,129]. On the contrary, some national versions seem to respond better to a trifactorial structure. For example, in the Thai [130], Iranian [74], Spanish [131], and Croatian versions [132], the results suggested that it would be better to consider a three-factor structure than a single-factor structure. It is likely that the observed differences were linked to the different linguistic, cultural, economic, and social characteristics of the samples studied.

Furthermore, cross-cultural differences may produce marked differences in nurses’ self-assessment of work ability. It has been observed that mean WAI score found in Israeli hospital nurses is relatively high, compared with that of European nurses [133]. Studies have also attempted to identify the extent to which individual health skills, in conjunction with sociodemographic influences, impact work ability. Workers’ literacy is of great importance. In a German study, the health literacy abilities accounted for 17.5% of the variance in the WAI score, while the extra sociodemographic background factors explained 27.5% of the variance [134]. The Thai authors [130] observed that workers’ poor knowledge of the concept of work ability tended to reduce the reliability of the questionnaire. Finally, the respondents’ self-awareness and ability to “look inside” themselves [135] could significantly vary the self-assessment.

In the absence of an external criterion to measure working capacity, we cannot say whether the low values reported by workers correspond to an actual lower capacity or whether they are the result of an excessively negative evaluation induced by occupational, emotional, or social factors. For example, in a previous study in which we used absences in the last three years as a measure of disability, we noticed that women reported lower WAI values than men but had the same number of absences [34]. This difference is an indication of different self-evaluation criteria in women and men, which could depend on work or emotional factors such as those studied here, or on other socio-cultural factors not yet studied.

Beyond its merit of having investigated simultaneously the contribution of occupation and emotional factors in the prediction of WA, this study also has some limitations. The main one is that it is a cross-sectional study, which therefore does not allow us to infer the direction of the relationships observed. Another limitation is the fact that it was carried out on workers from a single company, which limits the generalizability of the results to other working populations. However, the choice of a public health company makes our sample homogeneous with the 617,466 National Health Service employees in public facilities active in 2020 [136]. Cross-cultural differences, and uncontrolled variables which might be interfering with the accurate measurement of work ability, are another relevant limitation. Longitudinal multicenter studies will be more effective in clarifying the aspects reported by this study.

## 5. Conclusions

Occupational and emotional factors can influence employees’ self-assessment of their work ability. Work stress, but above all excessive commitment to work and intolerance towards possible unfavorable working conditions, can lead one to evaluate oneself as less capable at work. Similarly, anxious and depressed workers tend to rate themselves as less able, while workers who are happy and satisfied with their work tend to report high work ability. Age is associated with a decrease in the WAI component related to chronic pathologies that interfere with work, while the WAI component related to skills can increase with age and work experience. Knowledge of these associations is very useful for interpreting the results of surveys on work ability conducted in the workplace. Moreover, it can offer elements to improve work ability in health promotion programs.

The occupational physician could take advantage of the findings of this study, acting to improve the work ability of supervised workers. First, they can work in collaboration with management to promote the best placement of workers, taking into account their idiosyncrasy for certain working conditions (Work Annoyance) which has proven, together with Overcommitment, to be the main occupational predictor of poor work ability. A company policy aimed at discouraging overcommitment, by renouncing requests for overtime and off-time work and adopting non-intrusive leadership styles, would be recommended. To these corporate interventions, it would be useful to add campaigns to promote mental health and reduce work-related stress, with personalized interventions for workers. The company could benefit greatly from improving the working ability of its employees.

## Figures and Tables

**Table 1 healthcare-12-01731-t001:** Work ability in different categories of workers.

Category	WAI (Mean ± S.D.)
1. Physician (n = 98)	41.51 ± 5.42
2. Nurse (n = 321)	40.10 ± 5.68 ^a^
3. Technician (n = 33)	40.85 ± 4.99
4. Clerk (n = 32)	43.13 ± 3.77 ^b^

Note: Means with different letters were statistically different after Bonferroni correction.

**Table 2 healthcare-12-01731-t002:** Results of the principal component analysis on the WAI indicators (oblimin rotation with Kaiser normalization).

WAI Indicator	SEWA ^1^	IHRWA ^2^
1. Current work ability compared with lifetime best	0.798	0.170
2. Work ability in relation to the demands of the job	0.848	0.079
3. Number of current diseases diagnosed by a physician	0.197	0.748
4. Estimated work impairment due to illness	0.487	0.696
5. Sick leave during the past 12 months	−0.012	0.672
6. Personal prognosis of work ability 2 years from now	0.525	0.468
7. Mental resources	0.739	0.205

^1^ SEWA = subjectively estimated work ability; ^2^ IHRWA = ill-health-related work ability.

**Table 3 healthcare-12-01731-t003:** Bivariate correlations between the variables (Pearson’s r in the upper triangle, Spearman’s rho in the lower triangle).

	1	2	3	4	5	6	7	8	9	10	11	12	13
1. Age	1.000	−0.205 ***	0.226 ***	0.101 *	−0.031	0.041	−0.214 ***	−0.078	−0.110 **	−0.081	−0.106 **	0.028	−0.249 ***
2. Sex	−0.196 ***	1.000	0.063	−0.044	0.067	0.061	0.069	0.184 ***	0.190 ***	−0.075	−0.124 **	−0.096 **	−0.110 **
3. Work Annoyance	0.228 ***	0.079	1.000	0.308 ***	0.234 ***	−0.237 ***	−0.195 ***	0.308 ***	0.285 ***	−0.181 ***	−0.377 ***	−0.390 ***	−0.201 ***
4. ERI Stress	0.098 *	−0.030	0.325 ***	1.000	0.528 ***	−0.381 ***	−0.320 ***	0.423 ***	0.380 ***	−0.230 ***	−0.393 ***	−0.385 ***	−0.237 ***
5. Overcommitment	−0.030	0.092 *	0.210 ***	0.474 ***	1.000	−0.304 ***	−0.193 ***	0.538 ***	0.497 ***	−0.336 ***	−0.422 ***	−0.450 ***	−0.202 ***
6. Social Support	0.051	0.046	−0.218 ***	−0.389 ***	−0.310 ***	1.000	0.270 ***	−0.281 ***	−0.271 ***	0.194 ***	0.276 ***	0.317 ***	0.103 **
7. Satisfaction	−0.195 ***	0.071	−0.224 ***	−0.327 ***	−0.180 ***	0.271 ***	1.000	−0.203 ***	−0.205 ***	0.222 ***	0.258 ***	0.249 ***	0.163 ***
8. Anxiety	−0.057	0.191 ***	0.308 ***	0.390 ***	0.538 ***	−0.284 ***	−0.216 ***	1.000	0.773 ***	−0.410 ***	−0.490 ***	−0.481 ***	−0.295 ***
9. Depression	−0.083	0.203 ***	0.290 ***	0.364 ***	0.488 ***	−0.271 ***	−0.222 ***	0.786 ***	1.000	−0.419 ***	−0.490 ***	−0.520 ***	−0.240 ***
10. Happiness	−0.107 *	−0.065	−0.196 ***	−0.244 ***	−0.309 ***	0.217 ***	0.254 ***	−0.399 ***	−0.402 ***	1.000	0.396 ***	0.427 ***	0.188 ***
11. WAI	−0.106 *	−0.143 *	−0.369 ***	−0.376 ***	−0.376 ***	0.264 ***	0.237 ***	−0.488 ***	−0.464 ***	0.372 ***	1.000	0.878 ***	0.746 ***
12. SEWA ^1^	0.033	−0.110 **	−0.400 ***	−0.367 ***	−0.417 ***	0.325 ***	0.236 ***	−0.489 ***	−0.492 ***	0.392 ***	0.845 ***	1.000	0.336 ***
13. IHRWA ^2^	−0.247 ***	−0.119 **	−0.199 ***	−0.222 ***	−0.157 ***	0.063	0.161 ***	−0.268 ***	−0.226 ***	0.177 ***	0.693 ***	0.244 ***	1.000

***. The correlation is significant at the 0.001 level (2-tailed); **. The correlation is significant at the 0.01 level (2-tailed); *. The correlation is significant at the 0.05 level (2-tailed). ^1^ SEWA = subjectively estimated work ability; ^2^ IHRWA = ill-health-related work ability.

**Table 4 healthcare-12-01731-t004:** Effect of professional and emotional factors on the total score of the WAI and on that of SEWA and IHRWA components.

Variable	Model I			Model II		
	WAI Beta	SEWA ^1^	IHRWA ^2^	WAI Beta	SEWA ^1^	IHRWA ^2^
Age	#	0.097 *	−0.287 ***	−0.078 *	#	−0.303 ***
Sex	−0.094 *	#	−0.158 ***	#	#	−0.129 **
Work Annoyance	−0.269 **	−0.313 ***	#	−0.205 ***	−0.236 ***	#
ERI Stress	−0.112 *	#	−0.141 *	#	#	−0.103 ***
Overcommitment	−0.255 ***	−0.304 ***	−0.124 *	−0.139 **	−0.161 ***	#
Support	#	0.113 **	#	#	0.105 **	#
Satisfaction	0.133 **	0.123 **	#	0.083 *	#	#
Anxiety				−0.139 *	#	−0.243 ***
Depression				−0.150 *	−0.237 ***	#
Happiness				0.179 ***	0.214 ***	#
R^2^	0.293	0.327	0.141	0.374	0.412	0.179
VIF range ^3^	1.027–1.559	1.121–1.208	1.056–1.412	1.138–2.745	1.155–1.575	1.066–1.280

# = excluded during stepwise selection; ***: *p* < 0.001; **: *p* < 0.01; *: *p* < 0.05. Model I: age, sex, and occupational factors; Model II: age, sex, occupational factors, and emotional factors. ^1^ SEWA = subjectively estimated work ability”; ^2^ IHRWA = ill-health-related work ability. ^3^ VIF = variance inflation factor.

## Data Availability

Data are deposited on Zenodo (https://doi.org/10.5281/zenodo.12590363 (accessed on 28 August 2024).

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
