# Peer review of "Emotional and Work-Related Factors in the Self-Assessment of Work Ability among Italian Healthcare Workers"

_healthcare, 2024, doi:10.3390/healthcare12171731_

Round 1

Reviewer 1 Report

Comments and Suggestions for Authors

Comments and Suggestions for Authors.

  1. Clarity and Structure:

    • Abstract: The abstract provides a comprehensive overview, but consider tightening the language for conciseness. For example, the explanation of the WAI components could be more succinct.
    • Introduction: The introduction does a good job of contextualizing the study, but it could benefit from a clearer statement of the research gap. Explicitly stating how this study advances the current understanding of work ability assessment would strengthen the introduction.
    • Materials and Methods: The methodology is detailed, which is commendable. However, consider providing more information on the rationale behind the choice of specific statistical methods, such as the stepwise multiple hierarchical linear regression.
  2. Data Presentation:

    • The division of WAI into "subjectively estimated work ability" (SEWA) and "ill-health-related work ability" (IHRWA) is well-explained. However, when presenting results, ensure that the tables and figures are clearly labeled, and that legends provide sufficient detail to be understood independently of the main text.
    • Consider adding a flowchart to illustrate the participant selection process, including reasons for any exclusions. This would enhance the transparency of the study.
  3. Discussion:

    • The discussion effectively interprets the findings, but it could be enhanced by more explicitly comparing your results with those from similar studies. This would help situate your findings within the broader literature.
    • The limitations section is adequately addressed, but consider discussing potential biases introduced by self-report measures in more detail, particularly regarding the subjective nature of the WAI components.
  4. Literature Review:

    • The literature review is thorough but could benefit from the inclusion of more recent studies, particularly those published in the last two to three years. This would demonstrate the study's relevance to current research trends.
    • Some references are repeated or could be updated to more recent ones. Review the reference list for any outdated sources.
  5. Statistical Analysis:

    • The choice of a stepwise regression approach is justified; however, ensure that the criteria for including or excluding variables are clearly described in the methods section.
    • It would be beneficial to include a brief discussion on the potential for multicollinearity among predictor variables and how this was addressed.
  6. Ethical Considerations:

    • The study mentions ethics approval, which is good practice. However, consider expanding on how participant confidentiality was maintained, especially regarding the handling of sensitive health data.
  7. Language and Style:

    • The manuscript is generally well-written, but there are some areas where the language could be refined for clarity and precision. For example, avoid using passive voice excessively and ensure that all technical terms are clearly defined for the reader.
  8. Figures and Tables:

    • Ensure that all figures and tables are of high quality and are correctly referenced in the text. Descriptive titles and legends should be provided to allow them to stand alone.
  9. Conclusion:

    • The conclusion is succinct but could be strengthened by offering more concrete suggestions for future research. Additionally, reiterate the practical implications of your findings for occupational health practitioners.

Author Response

Reviewer #1

Comments and Suggestions for Authors.

  1. Clarity and Structure:
    • Abstract: The abstract provides a comprehensive overview, but consider tightening the language for conciseness. For example, the explanation of the WAI components could be more succinct.

R.: We thank the reviewer for the many useful suggestions he/she gave us to improve the article. We have happily adhered to all the indications. We would have gladly removed the definition of the two main components of the questionnaire from the abstract, but the presence of psychometric models with one, two, three or more components, which we report throughout the article, advised us against doing so in order not to make the abstract unclear for readers who are not aware of it. However, the abstract is composed of 202 words, therefore within the limits indicated by the Journal. We couldn't reduce it.

    • Introduction: The introduction does a good job of contextualizing the study, but it could benefit from a clearer statement of the research gap. Explicitly stating how this study advances the current understanding of work ability assessment would strengthen the introduction.

R.: We thank the reviewer for giving us the opportunity to explain the aims of the work in more detail. Accepting the reviewer's suggestion, which he/she also reiterated below, we have expanded the part of the introduction in which we explain the aims of the work and its originality with respect to the pre-existing literature.

    • Materials and Methods: The methodology is detailed, which is commendable. However, consider providing more information on the rationale behind the choice of specific statistical methods, such as the stepwise multiple hierarchical linear regression.

R.: It seemed logical to us to study the occupational factors first, because they are the ones that can potentially be changed within the company, and then the emotional ones, which would require individualized interventions on workers. We used a progressive variable selection method, to get an idea of the relative importance of the different factors. We have added this explanation in the text.

  1. Data Presentation:
    • The division of WAI into "subjectively estimated work ability" (SEWA) and "ill-health-related work ability" (IHRWA) is well-explained. However, when presenting results, ensure that the tables and figures are clearly labeled, and that legends provide sufficient detail to be understood independently of the main text.

R.: Thanks to the reviewer for reminding us to add this note to Table 3 and table 4.

    • Consider adding a flowchart to illustrate the participant selection process, including reasons for any exclusions. This would enhance the transparency of the study.

R.: All workers visited were invited to fill out the questionnaire with the information reported in this study. 91.6% of them participated. Among the reasons for non-participation, workers mainly indicated the lack of time to fill out the questionnaire; only two of them declared that they considered the health promotion activity useless. We added this information in the manuscript.

  1. Discussion:
    • The discussion effectively interprets the findings, but it could be enhanced by more explicitly comparing your results with those from similar studies. This would help situate your findings within the broader literature.

R.: Accepting the reviewer's invitation, we have broadened the comparison with the literature, introducing other references, both in the Introduction and in the Discussion. We have cited all the studies that have investigated the association of some of the occupational or emotional factors we studied with the WAI.

    • The limitations section is adequately addressed, but consider discussing potential biases introduced by self-report measures in more detail, particularly regarding the subjective nature of the WAI components.

R.: Following the reviewer's suggestions, we have inserted in the Discussion a paragraph dedicated to cross-cultural differences and individual variation in self-reporting, and we have mentioned these factors among the limitations.

  1. Literature Review:
    • The literature review is thorough but could benefit from the inclusion of more recent studies, particularly those published in the last two to three years. This would demonstrate the study's relevance to current research trends.

R.: We have firmly accepted the reviewer's recommendation. All articles on Work Ability published in 2023 and 2024 that were relevant to this study have been cited. In this version we have added some works that came out after the first version was submitted. The article has more than 130 references.

    • Some references are repeated or could be updated to more recent ones. Review the reference list for any outdated sources.

R.: The reviewer correctly noted that some authors were cited multiple times in the manuscript. However, the authors were cited twice when their relevant article contained various elements, which were referenced in separate parts of the manuscript. We have checked the list of references again and it seems that there are no articles that have lost their relevance.

  1. Statistical Analysis:
    • The choice of a stepwise regression approach is justified; however, ensure that the criteria for including or excluding variables are clearly described in the methods section.

R.: Accepting the advice, we have detailed in the methods both the mathematical selection criteria and the logical criteria for the inclusion of the predictors. We have also explained in the Introduction the reasons that led us to choose these statistical methods.

    • It would be beneficial to include a brief discussion on the potential for multicollinearity among predictor variables and how this was addressed.

R.: The reviewer highlighted a point of great importance in the study of regression. We tested the variables for multicollinearity using the variance inflation factor (VIF), which measures the correlation and strength of correlation between the predictor variables. The value for VIF starts at 1 and has no upper limit. A value between 1 and 5 indicates moderate correlation between a given independent variable and other independent variables in the model, but this is often not severe enough to require attention. In this study, in all tested models the VIF values of the variables were close to unity, indicating that the degree of correlation between the predictors in the model was not such as to influence the outcome. We have briefly explained the problem of multicollinearity and indicated the methods to control it. In Table 4 we added a row indicating the VIF range of the variables in each model. We have commented on these results in the manuscript.

  1. Ethical Considerations:
    • The study mentions ethics approval, which is good practice. However, consider expanding on how participant confidentiality was maintained, especially regarding the handling of sensitive health data.

R.: This point is also of utmost importance. We believe that the reader should be made aware of the exact conditions under which the study was conducted. Health surveillance of workers, which includes occupational risk prevention and health promotion activities, is mandatory in all workplaces where workers are exposed to occupational risks. The data collected in these activities are confidential and cannot be disclosed. The results relevant to health and safety are communicated in an anonymous collective form to the employer, the risk prevention service and the workers' representatives and may be the subject of scientific communications. We have added these explanations.

  1. Language and Style:
    • The manuscript is generally well-written, but there are some areas where the language could be refined for clarity and precision. For example, avoid using passive voice excessively and ensure that all technical terms are clearly defined for the reader.

R.: Thanks for the tip. The manuscript had several typos, and some terms needed more explanation. The use of the passive voice is frequent in Italian, much less so in English; we have tried to avoid this problem when possible, by explicitly indicating the subject of the action.

  1. Figures and Tables:
    • Ensure that all figures and tables are of high quality and are correctly referenced in the text. Descriptive titles and legends should be provided to allow them to stand alone.

R.: Thanks for the tip; we have checked the tables again. As mentioned above, we added notes to the tables when necessary.

  1. Conclusion:
    • The conclusion is succinct but could be strengthened by offering more concrete suggestions for future research. Additionally, reiterate the practical implications of your findings for occupational health practitioners.

R.: We added some considerations. The occupational physician could take advantage of the findings of this study, acting to improve the work ability of supervised workers. First, he/she can work in collaboration with management to promote the best placement of workers, taking into account their idiosyncrasy for certain working conditions (Work Annoyance) which has proven, together with Overcommitment, to be the main occupational predictor of poor work ability. A company policy aimed at discouraging overcommitment, by renouncing requests for overtime and off-time work and adopting non-intrusive leadership styles, would be recommended. To these corporate interventions it would be useful to add campaigns to promote mental health and reduce work-related stress, with personalized interventions on workers. The company could benefit greatly from improving the working ability of its employees.

We believe that this brief summary indicates the prospects of this study and its practical utility.

Reviewer 2 Report

Comments and Suggestions for Authors

Hi 

this is a neat study with a solid research method and appropriate analyses.

The implications are reasonable.

I do not have any major review to recommend, but mentioning in the limitation paragraph, the fact that other uncontrolled variables might be interfering with the accurate measurement of work-related distress, i.e., the respondents' self-awareness and ability to "look inside" themselves (see: 10.1080/15298868.2018.1543728)

Another minor revision: sometimes words are incorrectly broken (e.g., "de-pression" in the abstract).

Author Response

Reviewer #2

Hi 

this is a neat study with a solid research method and appropriate analyses.

The implications are reasonable.

I do not have any major review to recommend, but mentioning in the limitation paragraph, the fact that other uncontrolled variables might be interfering with the accurate measurement of work-related distress, i.e., the respondents' self-awareness and ability to "look inside" themselves (see: 10.1080/15298868.2018.1543728)

R.: We thank the reviewer for the time dedicated to our work. We absolutely agree that personal characteristics of the worker influence reporting; our study was born from this belief. Some features, such as the one listed here, have not been studied. We have added this statement in the Limitations section. We also cited the following paper: Aschieri, F.; Durosini, I.; Smith, J.D. Self-curiosity: Definition and measurement. Self and Identity 2020; 19:1, 105-115, DOI:10.1080/15298868.2018.1543728

Another minor revision: sometimes words are incorrectly broken (e.g., "de-pression" in the abstract).

R.: Thanks to the reviewer for reporting it. Unfortunately, the extra-hyphens were introduced by Word in the transfer of the file submitted for linguistic revision, and some of them had escaped our correction.

Reviewer 3 Report

Comments and Suggestions for Authors

The manuscript under review is devoted to the important and practically significant problem on factors of self-assessment of work abilities among health care workers (by example of workers from an Italian public health company). The purpose and content of the article correspond to scope of the journal Healthcare.

The authors conducted an interesting study using relevant methods and obtained valuable data. However, in my opinion, the presentation of the rationale and methods of the study in the text of the article need to be corrected.

I suggest that authors consider the following recommendations for improving the manuscript:

1)    Clarify and supplement the Title to more fully reflect the content of this study: “Emotional and work-related factors in the self-assessment of work ability among Italian health care workers

2)    The Introduction should provide a more detailed justification of the problem and the novelty of this study. Perhaps similar studies have not been conducted on Italian samples or on samples of health care workers? At the same time, lines 142-146 are better moved to Methods

3)    In the Methods section, you should explain why both Spearman and Pearson correlations were used. Was the data checked for normal distribution?

4)    Both in the Introduction and in the Discussion it is advisable to take into account possible cross-cultural differences between Italian health care workers and representatives of other countries and cultures. The cross-cultural aspect can also be mentioned in the limitations of the study and in the Conclusions

5)    Authors should carefully proofread the text and correct typos. For example, remove the extra hyphen in some words (‘de-pression’ at line 12, re-sources at line 65, etc.); add a description of WAI1 instead of ‘These are WAI1’ at line 50, etc.

Author Response

Reviewer #3

The manuscript under review is devoted to the important and practically significant problem on factors of self-assessment of work abilities among health care workers (by example of workers from an Italian public health company). The purpose and content of the article correspond to scope of the journal Healthcare.

The authors conducted an interesting study using relevant methods and obtained valuable data. However, in my opinion, the presentation of the rationale and methods of the study in the text of the article need to be corrected.

I suggest that authors consider the following recommendations for improving the manuscript:

Response: We sincerely thank the reviewer for the time he/she spent to examine our article and for the valuable suggestions he/she gave us to improve it.

  • Clarify and supplement the Titleto more fully reflect the content of this study: “Emotional and work-related factors in the self-assessment of work ability among Italian health care workers

Response: We gladly followed the reviewer's suggestion, which gave us the opportunity to better specify the scope of the study.

  • The Introductionshould provide a more detailed justification of the problem and the novelty of this study. Perhaps similar studies have not been conducted on Italian samples or on samples of health care workers? At the same time, lines 142-146 are better moved to Methods

R.: The reviewer is right. We tried to better explain what the gaps in the literature are and the reasons for this study. Although several studies have noted that WAI scores may be associated with work-related factors, there are no studies that have simultaneously assessed the influence of occupational and emotional factors on self-assessed workability. We have added this statement. As for the lines indicated, we promptly followed the suggestion, moving this section to the Methods.

  • In the Methodssection, you should explain why both Spearman and Pearson correlations were used. Was the data checked for normal distribution?

R.: We thank the reviewer for this observation, we promptly accepted the request to better explain the logical process that led us to the use of non-parametric and parametric methods. The distribution of the relevant variables was examined using the Shapiro-Wilk and Kolmogorov-Smirnova tests to see whether it was normal. We followed the general principle of applying nonparametric tests to non-normally distributed variables; however, as reported by Lumley et al. [ref], the assumption of normality is only required for small samples, due to the central limit theorem. With sample sizes exceeding 30, as it is the case here, violations of the normality assumptions are not problematic, and the use of both parametric and non-parametric tests is appropriate. We have added this note to the text.

Lumley, T., Diehr, P., Emerson, S., & Chen, L. (2002). The importance of the normality assumption in large public health data sets. Annual Review of Public Health, 23(1), 151–169. https://doi.org/10.1146/annurev.publhealth.23.100901.140546

  • Both in the Introductionand in the Discussion it is advisable to take into account possible cross-cultural differences between Italian health care workers and representatives of other countries and cultures. The cross-cultural aspect can also be mentioned in the limitations of the study and in the Conclusions

R.: We thank the reviewer because this observation allowed us to address in greater detail a theme already present in the Introduction and which motivated our study on the variability of the WAI. Cultural and social factors most likely influence workers' self-assessments. An indirect demonstration of such differences comes from the variation in psychometric assessments. Most of the authors confirmed the unifactorial structure suggested by the creators of the questionnaire [ref] or, while noting that an exploratory analysis of the principal components supported validity of the one-factor structure, suggested better properties in relation to a two-factor structure supported by confirmatory analyses [ref]. On the contrary, some national versions seem to respond better to a trifactorial structure. For example, in the Thai [ref], Iranian [ref], Spanish [ref], and Croatian versions [ref] the results suggest that it would be better to consider a three-factor structure than a single-factor structure. Furthermore, cross-cultural differences may produce marked differences in nurses' self-assessment of work ability. It has been observed that mean WAI score found in Israeli hospital nurses is relatively high as compared with that of European nurses [ref]. Studies have also attempted to identify the extent to which individual health skills, in conjunction with sociodemographic influences, impact work ability. In a German study, the health literacy abilities accounted for 17.5% of the variance in the WAI score, while the extra sociodemographic background factors explained 27.5% of the variance [ref]. Finally, the Thai authors [ref] observed that workers' poor knowledge of the concept of work ability tends to reduce the reliability of the questionnaire. We have included these concepts in the Discussion, briefly mentioning the problem in the Introduction.

5)    Authors should carefully proofread the text and correct typos. For example, remove the extra hyphen in some words (‘de-pression’ at line 12, re-sources at line 65, etc.); add a description of WAI1 instead of ‘These are WAI1’ at line 50, etc.

R.: Thanks to the reviewer for reporting it. Unfortunately, the extra-hyphens were introduced by Word in the transfer of the file submitted for linguistic revision, and some of them had escaped our correction. On line 50, we inserted the sentence: “The seven components are characterized as follows:…”